# The Top 50 Most Cited Articles on Special Olympics: A Bibliometric Analysis

**DOI:** 10.3390/ijerph191610150

**Published:** 2022-08-16

**Authors:** Selina Khoo, Payam Ansari, Jacob John, Mark Brooke

**Affiliations:** 1Centre for Sport and Exercice Sciences, Universiti Malaya, Kuala Lumpur 50603, Malaysia; 2Disability Sport Research Centre, Universiti Malaya, Kuala Lumpur 50603, Malaysia; 3DCU Business School, Dublin City University, D09Y5NO Dublin, Ireland; 4Department of Restorative Dentistry, Faculty of Dentistry, Universiti Malaya, Kuala Lumpur 50603, Malaysia; 5Centre for English Language Communication, National University of Singapore, Singapore 117511, Singapore

**Keywords:** disability sport, intellectual disability, citation, authorship, scientometric

## Abstract

The Special Olympics was established in 1968 to “provide year-round sports training and athletic competition in a variety of Olympic-type sports for children and adults with intellectual disabilities”. It has gained recognition in the field of sports and healthcare of persons with intellectual disability, with a large number of dedicated researchers and institutions all over the world. However, there is an urgent need to analyze the progress and current status of this research field to identify knowledge gaps and develop this discipline. The aim of this study was to analyze the scientific production of the Special Olympics and report the bibliometric characteristics of the top 50 most cited Special Olympics publications. A systematic search was conducted on the Scopus database and bibliometric data were extracted and analyzed. The top 50 publications received 1632 citations. A total of 138 authors (63 female and 75 male) contributed to these publications. The two main areas of study were the physical health of Special Olympics athletes (*n* = 27) and the psycho-social health of athletes (*n* = 12).

## 1. Introduction

Physical activity has been linked to physical, psychological, social, and emotional benefits for persons with intellectual disability [1,2,3]. The World Health Organization recommends that adults with disabilities participate in at least 150–300 min of moderate-intensity aerobic activity each week, or at least 75–150 min of vigorous-intensity aerobic activity, or an equivalent combination of both to gain health benefits [4]. For children and adolescents with disabilities, the recommendation is an average of 60 min of moderate-to-vigorous intensity physical activity a week. Despite the well-established benefits of physical activity, it is estimated that 27.5% of adults [5] and 81% of adolescents [6] do not get enough physical activity. Compared to persons without disabilities, more persons with intellectual disabilities do not meet the physical activity recommendations. A systematic review by Dairo et al. [7] reported that only 9% of adults with intellectual disability met the physical activity guidelines.

Physical activity refers to any bodily movement employing the skeletal muscles and resulting in energy expenditure [8]. Sport, a subset of physical activity, refers to activities which are physical, competitive, institutionalized, as well as motivated by internal and external rewards [9]. As there is limited opportunity for persons with intellectual disability to participate in sport, the Special Olympics was established in the United States in 1968 “to provide year-round sports training and athletic competition in a variety of Olympic-type sports for children and adults with intellectual disabilities, giving them and participate in a sharing of gifts, skills and friendship with their families, other Special Olympics athletes and the community” [10]. From the first International Special Olympics Games in Chicago in 1968, the Special Olympics now supports more than 5 million athletes, 500,000 coaches, 1 million volunteers, and 100,000 competitions through its programs in 200 countries [11]. It has grown to become an established institution that unites people in a shared belief of a more just and welcoming world. The popularity of this movement is attributed to its success in helping athletes with an intellectual disability improve their physical fitness, general health and motor skills, self-confidence, social competence, self- image, and develop and maintain friendships [12]. Special Olympics participation has been found to have positive physical, psychological/emotional, and social outcomes [13].

In addition to providing the opportunity for athletes with intellectual disability to participate in sports, the Special Olympics has also created opportunities for family, care givers, coaches, community members, local leaders, and others to join forces to change attitudes and the wellbeing of persons with intellectual disability [10]. Special Olympics was said to have also been instrumental at promoting social inclusion and leadership qualities among people of various abilities to positively impact their communities [14]. Over the years, there have been many impactful studies on all the benefits and even concerns with regards to the Special Olympics and its service delivery reported from the viewpoint of athletes, parents, coaches, volunteers, media, and many governmental and non- governmental agencies [15].

The Special Olympics has gained recognition in the field of sports and healthcare of persons with intellectual disability, with a large number of dedicated researchers and institutions all over the world. There is an urgent need to analyze the scientific production of scientists, research units, institutions, or countries by taking into consideration the historical development of this discipline or by quantifying its role in the domain of disability sport and healthcare. One way to map the development of a field is to conduct bibliometric analyses on the available literature of that field. As González-Serrano et al. [16] stated, by using bibliographic indexes, the most significant literature of specific research fields are analyzed through bibliometric studies. These analyses help to learn the past trends of a field, and understand the progress of the investigations, while allowing the future development of research via its indicators [17].

To date, Khoo et al.’s [18] bibliometric study remains the most recent on disability sport. These authors conducted a systematic search of the Web of Science Core Collection citation index on 26 June 2017, for publications published since 1980, to identify the top 50 most cited publications in disability sport. They found that the majority of the publications can be categorized as sociological and psychological as well as training and competition effects. The most researched events were the Paralympics and Special Olympics. However, they found only five from their database focusing on the Special Olympics related to social competence, self-concept, obesity, and epidemiology. The main aim of this bibliometric review is to solely analyze the scientific production of the Special Olympics, which is the largest sports organization for children and adults with intellectual disability globally. To achieve this aim, we identify key researchers and research areas in the Special Olympics, summarize the extent, range, and nature of these research topics, and identify gaps in the literature.

## 2. Materials and Methods

We conducted a systematic search on 4 March 2022, in Scopus for publications with “Special Olympic*” in the title, abstract, or keywords. Scopus is the largest database of multidisciplinary publications in the social sciences [19]. The inclusion criteria were as follows:Research involving Special Olympics athletes, family, volunteers, coaches, managers, and spectators or conducted at Special Olympics events and training.Media coverage, sponsorship, or management of Special Olympics events.Administration, governance, and organization of Special Olympics organizations.Policy papers citing Special Olympics as their basis or as a contributor.

The search yielded 422 results. The search results were extracted, then imported to an Excel file, where the rest of the analyses were conducted. After excluding books, editorial, letters, notes, and short surveys (*n* = 19), the remaining 403 publications comprised journal articles, reviews, conference proceedings, and book chapters. We included these types of publications because they are peer-reviewed and citable. This follows Shilbury [20] who also included only “*citable items*”, excluding notes, editorials, and letters in his “A bibliometric analysis of four sport management journals”. The conference papers and book chapters in our data set received zero to seven citations. In the next stage, two authors (S.K. and M.B.) screened titles and abstracts against the inclusion criteria. Disagreements were resolved by a third author (P.A). After excluding 70 publications, we were left with 333 publications which received 3227 citations in total.

We analyzed the citation structure to draw a general trend and yearly structure of the Special Olympics literature. Two rounds of analyses were conducted, first for the entire data set and then on the top 50 most cited publications. We identified the top 50 most cited publications in order to evaluate the impact of the research and map out the evolution of the field based on the most influential publications. This is a common practice, and there are 28 bibliometric analyses of the “Top 50” cited studies published in different fields and topics available in Scopus, including hip arthroscopy [21], carpal tunnel syndrome [22], and respiratory system [23]. We performed authorship and keywords analyses for the entire data set before reviewing the top 50 most cited publications. The following section shows the results for the 333 publications.

## 3. Results

### 3.1. Overall Trend

From the first publication on the Special Olympics in 1978 (by Polloway and Smith) [24], there were less than 10 publications per year until 2007, with a few years with no publications. From 2011, there were more than 15 publications a year. By 2015, publications had increased by 100% from 10 in 2010 to 20. About half (53.7%) of the publications were published in the last 10 years with the most published in 2019 and 2020 (22 publications per year). These increases in publication numbers could demonstrate a consistently growing interest in this field of research. The number of publications is shown in Figure 1.

Of the 333 publications, most were journal articles (91.6%) and published in the English language (97%). There were also publications in Croatian (*n* = 1), German (*n* = 2), Polish (*n* = 6), and English/Polish (*n* = 1).

### 3.2. Citation Analysis

About three quarters (78.9%) of the publications have been cited and 30 (9%) publications have received only one citation. Those published in 2003 received the greatest number of citations with an average 32.37 citations for the eight cited publications that year. The most cited article in the corpus with 71 citations is from Dykens and Cohen published in the Journal of the American Academy of Child and Adolescent Psychiatry in 1996 entitled “Effects of Special Olympics International on social competence in persons with mental retardation” [25].

### 3.3. Authorship Analysis

A total of 802 authors contributed to the Special Olympics publications between 1978 and 4 March 2022. We considered the number of publications for the most prolific authors. V. A. Temple from the University of Victoria, Canada was the most prolific author with 16 publications (please see Table 1). Only four other authors (M. Lloyd, S.P. Perlman, L. Marks, and J.T. Foley) have more than 10 publications each. Top authors V.A. Temple, M. Lloyd, and J.T. Foley collaborated considerably from 2012 to 2018 to have the highest number of co-authorships (*n* = 11). The majority (81.5%) of the authors only contributed one publication on the Special Olympics. Most of the published works were produced by three or more authors. There were fewer contributions by two co-authors (*n* = 42), or by single authors (*n* = 33).

The universities with the most number of publications were the affiliations of the most prolific authors. The top three most prolific universities are the University of Victoria, Canada; University of Ontario, Canada; and Ghent University, Belgium.

Co-authorship analysis was performed using VOSviewer software (developed by the Centre for Science and Technology Studies at the University of Leiden, The Netherlands) to identify the authors’ network (see Figure 2). There are five clusters. There are seven authors in the red cluster with the most prolific author being Marks, L. The publications of that cluster are related to the oral health of Special Olympics athletes. The five authors in the green cluster are involved in international studies on the health of persons with intellectual disabilities. The four members of the blue cluster, with the most prolific author being McConkey, R, have publications related to social inclusion. The yellow cluster, which includes the two most prolific authors (Temple, V. A. and Lloyd, M.), has published research on the physical health and fitness of Special Olympics athletes. The purple cluster has only three authors and their publications are related to the experiences of Special Olympics athletes.

### 3.4. Most Influential Authors

We considered citation count for the most influential authors. V. A. Temple is the author with the most citations (*n* = 192) with an average of 12 citations per publication. The two authors (M. Lloyd and J. T. Foley) who co-published 11 works with V. A. Temple were also highly cited with more than 100 citations (*n* = 146; *n* = 143, respectively). S. P. Perlman is the author with the second highest number of citations (*n* = 163). Interestingly, despite only producing two publications, E. M. Dykens has more than 100 citations (*n* = 136). Y. Hutzer has 110 citations from his four publications. Please see Table 2.

A total of 43 countries contributed to the publications, with the United States being by far the most productive country with 170 publications. Canada and the UK are the second highest contributors at 39 and 37, followed by Belgium with 21. Eleven countries contributed only one publication each. There were publications from all the regions of the world. In additional to Belgium and the UK, 20 other European nations contributed, with the majority except Croatia, France, Georgia, and Switzerland (*n* = 1) producing at least two per nation. The most prolific Asian country is Israel (*n* = 7). Other Asian countries contributed publications at four per nation (India and Turkey), two per nation (China, Hong Kong, Indonesia, Malaysia, South Korea, Taiwan, and United Arab Emirates), or one per nation (Kazakhstan, Pakistan, Saudi Arabia, and Singapore). Australia (*n* = 13) and New Zealand (*n* = 2) were the only two countries from Oceania whereas Puerto Rico (*n* = 2) and Brazil (*n* = 1) represented the Americas. Kenya (*n* = 2) and Nigeria (*n* = 1) were the sole African nations to contribute.

### 3.5. Authors’ Keywords Analysis

We conducted an analysis of keywords that authors used in their publications. Ninety keywords appeared at least twice. Unsurprisingly, the top keyword was Special Olympics, which was found in 94 publications. The second most frequent keywords were intellectual disability (*n* = 79) and intellectual disabilities (*n* = 26). Related terms, Down syndrome and mental retardation, occurred in 10 publications each. As the term intellectual disability has replaced mental retardation, the last time mental retardation was used as a keyword was in 2010. Other frequent keywords were related to health (oral health, obesity, overweight, body mass index), physical activity and sports (sport, physical activity, Unified sports, physical fitness), and people (adults, youth, children, volunteer). Inclusion was used as a keyword in nine publications. The Special Olympics launched Unified Sports in 1988 [10]. This program promotes inclusion as persons with and without intellectual disabilities play on the same team. The Special Olympics started the Healthy Athlete program in 1997. This program provides free health screenings and education to Special Olympics athletes for physical examination, vision health, auditory, dentistry, nutrition, emotional health, physical therapy, and podiatry. This might explain the frequently used keywords related to health in the publications. Please see Table 3.

### 3.6. Most Popular Journals

A total of 164 publications have been published related to the Special Olympics, with *Adapted Physical Activity Quarterly* and *Journal of Intellectual Disability Research* publishing the most at 16 each. Other journals have published subject matter related to disability, special care dentistry, sport, or rehabilitation. Publications in *Adapted Physical Activity Quarterly* have the highest number of citations (*n* = 469), making the average citation per publication 29.31. Many of these journals have a long history and began publishing soon after 1978 when Polloway and Smith published the first Special Olympics research (*Adapted Physical Activity Quarterly*, 1984 publishing four issues per year, and the *Journal of Intellectual Disability Research*, from 1957 with up to 12 issues per volume over the last 15 years). Other journals which have published the most Special Olympics research have also been active for decades (*Research in Developmental Disabilities*, 1987; the *Journal of Applied Research in Intellectual Disabilities*, 1988, and *Special Care in Dentistry*, 1981). Please see Table 4.

Only 50 of the publications acknowledged funding with the Centers for Disease Control and Prevention in the United States (*n* = 12) and Special Olympics (*n* = 8) funding the most publications. Sixteen agencies funded only one publication each. This could be due to a lack of money or because funding agencies do not tend to specialize in research on the Special Olympics.

### 3.7. Top 50 Most Cited Publications

The top 50 most cited publications related to the Special Olympics are shown in Appendix A. They were published in the English language between 1989 and 2019 and were cited between 21 and 71 times. The two most cited publications with 71 and 69 citations are Dykens and Cohen’s (1996) article entitled “Effects of Special Olympics International on social competence in persons with mental retardation” published in the *Journal of the American Academy of Child and Adolescent Psychiatry* [24] and Weiss et al.’s (2003) “Involvement in Special Olympics and its relations to self-concept and actual competency in participants with developmental disabilities” in *Research in Developmental Disabilities* [25]. Ozer et al.’s (2012) article “Effects of a Special Olympics Unified Sports soccer program on psycho-social attributes of youth with and without intellectual disability” in *Research in Developmental Disabilities* [26], Pezzementi and Fisher’s (2005) “Oral health status of people with intellectual disabilities in the southeastern United States” in the *Journal of the American Dental Association* [27], and Dykens et al.’s (1998) “Exercise and sports in children and adolescents with developmental disabilities: Positive physical and psychosocial effects” in the journal *Child and Adolescent Psychiatric Clinics of North America* [28] received 65 citations each. After these articles, the citation numbers drop first to 60, then 52, and decline quite sharply to below 40 for the 16th highest cited article.

A total of 138 authors contributed to these three reviews and 47 articles. The publication with the highest citation is “Effects of Special Olympics International on social competence in persons with mental retardation” [24]. V. A. Temple is also the most prolific author, contributing four publications. Three researchers (J. T. Foley, M. Lloyd, and K. Storey) co-authored three articles, 24 authors contributed two, while most (*n* = 110) contributed only one publication. There were only eight single-authored publications in the list. K. Storey has two single-authored publications. The publication with the highest average citation per year of nine is “Oral health of adults with intellectual disabilities: a systematic review” by Ward et al. [29]. The second and third most cited per year were from McConkey, Dowling, Hassan, and Menke [14] at 6.67, followed by Özer, Baran, Aktop, Nalbant, Ağlamış, and Hutzler [26] and Wu et al. [30] at 6.5. The lowest average citation per year from the top 50 most cited was by Roper [31] in 1990 at 0.78. Nearly half (45%) of the publications were produced in *Adapted Physical Activity Quarterly* (*n* = 10), *Journal of Intellectual Disability Research* (*n* = 7), and *Research in Developmental Disabilities* (*n* = 7).

There were a total number of 138 authors who produced the most cited 50 publications. Of these, 63 were female and 75 were male contributors. The most prolific author, V. A. Temple with four publications, is female. The next three most prolific, with three publications each, were male (J. T. Foley; K. Story) and female (M. Lloyd). Of the 24 authors who had two publications each, 16 were female and eight, male. Of the 110 authors with one publication, 57 were female, and 53, male. Regarding the most cited 50 publications, the two main areas of study were the physical health of Special Olympics athletes (*n* = 24) and the psycho-social health of athletes (*n* = 8). Two publications highlighted both physical and psycho-social health [13,26]. Other subject matter included public attitudes towards persons with intellectual disabilities through the Special Olympics (*n* = 3) [27,28,29], retaining volunteers and volunteer motivation (*n* = 3) [18,30,31], the pros and cons of the Special Olympics (*n* = 2) [32,33], social inclusion (*n* = 2) [14,34], and motives for participating in Special Olympics (*n* = 2) [35,36].

For the first main area, physical health, there were predominantly articles on physical fitness (*n* = 5) [37,38,39,40,41] and oral health (*n* = 8) [42,43,44,45,46,47,48,49]. Additionally, articles examined body mass index and risk of obesity (*n* = 4) [50,51,52,53], eye (*n* = 3) [54,55,56], hearing (*n* = 2) [57,58], and spine health (*n* = 1) [59], physical activity and sedentary behavior (*n* = 1) [60]. For psychosocial health, the publications covered subject matter such as Special Olympics athletes’ self-esteem and self-worth, social competence, perceived competence in sport as well as Special Olympics athletes’ attitudes towards winning.

Eleven of the top 50 studies included more than 500 participants; the two largest scale projects included approximately 12,000 participants in each. One of the large-scale projects utilized body mass index records from the Special Olympics International Health Promotion databases [52], whereas the other assessed the oral health status of Special Olympics athletes in the United States [42]. Most (*n* = 17) studies involved between 50 to 500 participants, and seven were small scale studies with less than 50 participants. The two smallest studies only included 12 volunteers and 13 Special Olympics athletes. Both children and adults were selected as participants in the studies.

As many of these studies are related to the physical health of Special Olympics athletes, the majority present data from medical examinations such as reports of screenings as part of the Special Smiles program for oral health, or other tests for cardiovascular fitness, eyes, and hearing health, as well as body mass index measurements. Apart from that, there were a number of quantitative studies employing questionnaire scales (*n* = 9) on psychological subject matter such as the Self-Esteem Inventory, Sport Motivation Questionnaire, the Friendship Activity Scale, Perceived Competence Scale for Children, the Adjective Checklist, as well as behavioral content such as the Children Behavior Checklist. These studies were commonly exploring participants at Special Olympics Games or training; four explored participants of Unified Sports. A small number of studies (*n* = 5) involved interviews with Special Olympians, coaches, parents, and community leaders. Nine studies were comparatives between Special Olympics and non-Special Olympics athletes with developmental disabilities; three compared active Special Olympics vs. sedentary non-Special Olympics with developmental disabilities; and some articles (*n* = 7) were reviews of available data from public sites such as the Special Olympics International Health Promotion database, or reviews of the research literature on the Special Olympics to explore matters such as the pros and cons of the Games and suggestions for future directions.

## 4. Discussion

The aim of this study was to analyze the scientific production of the Special Olympics and report the bibliometric characteristics of the top 50 most cited SO publications. Based on this analysis, we found an increasing trend in the number of publications on Special Olympics. This appears to be in line with the growth of the Special Olympics movement. Special Olympics, which began in the United States, started to spread to other countries in the 1970s. In 2014, there were more than 4 million Special Olympics athletes around the world and in 2021, the number was nearly 5.8 million [10]. It can be observed that significant peaks in research occur in 1983, 1990, 1993, 1996, 1998, 2001, and 2003. After that, 2015 and 2020 can be highlighted. Interestingly, these dates correspond to the year of a Special Olympics World Summer Games and the year just before or following the Games. In 2003, the Games were held outside the United States for the first time, in Dublin, Ireland. In 2007, they were held in China; and in 2011, in Greece. Then in 2015, the Games took place in Los Angeles, which represents a strong increase. After that, in line with this trend, the Games were held in Abu Dhabi in 2019.

Except for the United States which have hosted both the Special Olympics Summer and Winter World Games and Canada which have hosted the Winter World Games, other countries which have hosted the Games were not among those with the highest number of publications. It is possible that the number of publications correlates with the extent to which a country is increasingly recognizing the importance of making people with intellectual disabilities visible rather than this demographic group remaining marginalized and underrepresented in our societies. However, even in the United States, people with intellectual disabilities often remain excluded in health research due to challenges with research access, recruitment, and retention [61].

The top 50 most cited publications related to Special Olympics were cited between 21 and 71 times. This was less than the citations of the top 50 most cited publications in disability sport which were cited between 26 and 126 times [18]. The two main areas of research on the Special Olympics were the physical health and the psycho-social health of athletes. Very little research has explored media coverage, sponsorship, and management of Special Olympics events or administration, governance, and organization of Special Olympics organizations. There is also limited research on competitivity and successes in performance, which is common for Paralympic research [18]. Only physical fitness and perceived competence in sport were found. This differs considerably with the literature on Paralympic athletes. For example, in their systematic review, Jefferies, Gallagher, and Dunne [62] examined five key databases (CINAHL Plus from 1937 to April 2012), ISI Web of Science (sciexpanded, SSCI, A&HCI; from 1945 to April 2012), MEDLINE (via OvidSP; from 1946 to April 2012), PsycInfo (from 1919 to April 2012), and SportDiscus (from 1800 to April 2012) that index relevant literature on the Paralympics. They identified a number of subject areas, including ‘participation, motivations, and goals’; ‘stress and coping’; ‘knowledge and attitudes towards doping’; and ‘transitions to retirement’. They conclude that the literature has a strong focus on athletes engaging in Paralympic sport for accomplishment and prowess were key indicators in the research literature. Similarly, in Khoo et al.’s [18] bibliometric analysis of the top 50 most cited publications in disability sport, the Paralympics was the most studied sport event, and the top-cited publications were categorized under ‘training’ and ‘competition effects’. In our study, we found very little research covering subject matter related to performance and competitivity in the Special Olympics. Rather, it concerned physical and mental health, families, and caregivers. This is in line with the mission of the Special Olympics to provide year-round training and competition for persons with intellectual disability as well as opportunities to develop physical fitness, friendship, and social inclusion. Performance and competitivity are therefore not the main aim of the Special Olympics.

Future research directions may include media coverage, sponsorship, as well as administration and management of the Special Olympics. As the Special Olympics include a variety of sports, future research could examine associations between the different sports and various health parameters. Coaches are very important in Special Olympics and future research could determine their needs in order to better equip them to train Special Olympics athletes. Future research could also consider involving Special Olympics athletes in informing and shaping the research. Special Olympics is a global program and offers the unique opportunity for comparative studies across regions and countries to determine the differences in implementation and effectiveness.

## 5. Conclusions

The main aim of this bibliometric review was to analyze the scientific production of the Special Olympics, which is the largest sports organization for children and adults with intellectual disability globally. The Special Olympics is a relatively new area of research which has gained popularity in the last 10 years. It is comparatively under-researched, with only 333 publications from 1978. Authors from North America have mainly contributed to the research. There has also been limited funding for Special Olympics research. An analysis of the top 50 most cited publications in the area showed that the physical health and psycho-social health of Special Olympics athletes were the main areas of research. There are gaps in the literature on Special Olympics, with many opportunities for research, including in media coverage, sponsorship, administration and management of the Special Olympics, as well as comparative studies of Special Olympics sports, countries, and regions.

## Figures and Tables

**Figure 1 ijerph-19-10150-f001:**
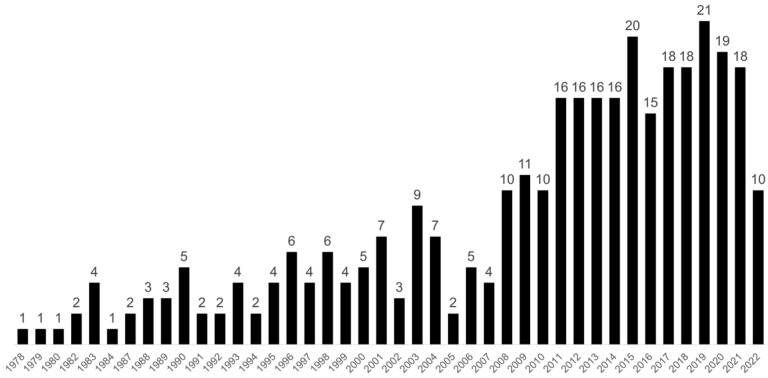
Special Olympics publications between 1978 and 4 March 2022.

**Figure 2 ijerph-19-10150-f002:**
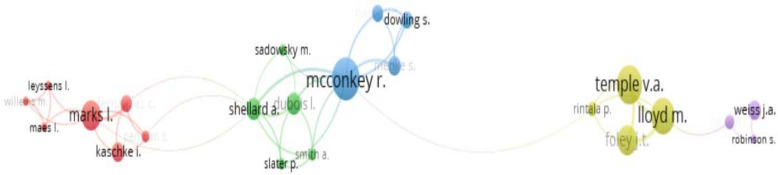
Authors network.

**Table 1 ijerph-19-10150-t001:** The top 10 most prolific authors of Special Olympics publications between 1978 and 4 March 2022.

Author	Institution	Country	TP	TC
Temple, V. A.	University of Victoria	Canada	16	192
Lloyd, M.	University of Ontario	Canada	15	146
Perlman, S. P.	Boston University	United States	13	163
Marks, L.	Ghent University Hospital	Belgium	13	117
Foley, J. T.	State University of New York College, Cortland,	United States	12	143
McConkey, R.	Special Olympics International	United States	10	29
Weiss, J. A.	York University	Canada	8	159
Fernandez C.	Ghent University	Belgium	8	69
Shellard, A.	Special Olympics International	United States	7	24
Dubois, L.	Special Olympics International	United States	7	11

This table shows the top 10 most prolific authors who contributed to the Special Olympics literature between 1978 and 4 March 2022. It also lists the total number of publications by each author (TP) and total citations associated with these publications (TC).

**Table 2 ijerph-19-10150-t002:** The top 10 most influential authors between 1978 and 4 March 2022.

Authors	Institution	Country	TC	TP
Temple, V. A.	University of Victoria	Canada	192	16
Perlman, S. P.	Boston University	United States	163	13
Weiss, J. A.	York University	Canada	159	8
Lloyd, M.	University of Ontario	Canada	146	15
Foley, J. T.	State University of New York College, Cortland,	United States	143	12
Dykens, E. M.	University of California	United States	136	2
Marks, L.	Ghent University Hospital	Belgium	117	13
Hutzler, Y.	Wingate Institute	Israel	110	4
Dowling, S.	University of Ulster	United Kingdom	81	4
Hassan, D.	University of Ulster	United Kingdom	81	4

TC: Total citations; TP: Total publications.

**Table 3 ijerph-19-10150-t003:** Top 20 keywords in Special Olympics publications between 1978 and 4 March 2022.

Keyword	*n*	Keyword	*n*
Special Olympics	94	Overweight	9
Intellectual disability	79	Body mass index	8
Intellectual disabilities	26	Adults	7
Oral health	19	physical activity	7
obesity	14	Sports	7
sport	13	Youth	7
disability	10	Children	6
Down syndrome	10	Athletes	5
Mental retardation	10	BMI	5
Inclusion	9	developmental disabilities	5

**Table 4 ijerph-19-10150-t004:** Top 10 most prolific Scopus-indexed journals publishing Special Olympics publications between 1978 and 4 March 2022.

Journal	TP	TC	C/P
*Adapted Physical Activity Quarterly*	16	469	29.31
*Journal of Intellectual Disability Research*	16	260	16.25
*Research in Developmental Disabilities*	13	326	25.08
*Journal of Applied Research in Intellectual Disabilities*	12	89	7.42
*Special Care in Dentistry*	11	150	13.64
*Sport in Society*	8	52	6.50
*European Journal of Adapted Physical Activity*	6	12	2
*Journal of Disability and Religion*	6	7	1.17
*Journal of Intellectual and Developmental Disability*	6	51	8.50
*Journal of Policy and Practice in Intellectual Disabilities*	6	30	5
*Journal of Physical Education and Sport*	5	33	6.60
*Postepy Rehabilitacji*	5	4	0.80

TP: total publications; TC: total citations; C/P: citation per publication.

## Data Availability

No new data were created or analyzed in this study. Data sharing is not applicable to this article.

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
