# Peer review of "The Top 50 Most Cited Articles on Special Olympics: A Bibliometric Analysis"

_ijerph, 2022, doi:10.3390/ijerph191610150_

Round 1

Reviewer 1 Report

Dear authors,

I’m pleased with the changes that you made. I believe that manuscript can be accepted in its present form.

Best regards

Author Response

Thank you.

Reviewer 2 Report

The authors made considerable changes to improve the manuscript. 

However, some of the previous comments were not addressed. 

Specifically, comments 2, 3e, and 4.

- (2) The use of 50 top cited studies should be justified, why not use 333?

- (3e) "The TOP 50 most cited articles/publications should be presented in a table including the indicators presented in text, e.g. citation count, study area, country, etc. Also, these publications should be cited in text."

- (4) "The main purpose of a review article is to critically analyze the literature, identify the gap, and provide directions for future research. All of these are missing in this paper." These should be discussed and not only mentioned in 1 sentence.

Additionally, from L371-374: " In our study, we found very little research covering subject matter related to performance and competitivity in the Special Olympics. Rather, it concerned physical and mental health, families and caregivers. This is a significant gap in the literature on Special Olympics with many opportunities for research."

- Discuss why "performance and competitivity in the Special Olympics" are important topics to study?

- Discuss broadly what are the "many opportunities for research"

Author Response

This manuscript is a resubmission of an earlier submission. The following is a list of the peer review reports and author responses from that submission.

Round 1

Reviewer 1 Report

I think more statistichal analisis can be done, so that the conclusions can be more interesting

Reviewer 2 Report

The paper analyzes the scientific works on Special Olympics using bibliometric method. The results of the paper are interesting and the topic fits well with the aims of special issue "Application of Bibliometrics in Health Research". However, the paper lacks scientific value, hence, it cannot be accepted at the current state. Specific comments are as follows:

1. The Introduction presented a good motivation and research background. However, it failed to present the state-of-art, particularly what reviews on Special Olympics have already been done to date. From this, the literature gap should be identified to justify the proposed academic contribution of the paper.

2. The Methodology should be discussed in detail. A flowchart of how 422 publications deduced to 333 should be presented. The use of 50 top publications should be justified.

3. The results of the bibliometric analysis are well-presented. However, the results lack critical analysis and discussion.

a. For instance, there are peaks in Figure 1. Are these not related to when the Special Olympics are held?

b. Section 3.4 is entitled "Most Influential Authors", yet the results tackle the most cited publications. Also, how come that in L161, "After these articles, the citation numbers drop first to 60, then 52 and decline quite sharply to below 40 for the 16th highest cited article" citations drop or decline you are referring to different publications?

c. What is the definition of "prolific" and "influential"? Both Tables 1 and 2 presented authors with the most number of citations and publications with C/P. It seems the "prolific" refers to the authors with most number of publications, while "influential" for most number of citations. Then what about the C/P?

d. Country analysis should also be done and compare where the Olympics was held.

e. The TOP 50 most cited articles/publications should be presented in a table including the indicators presented in text, e.g. citation count, study area, country, etc. Also, these publications should be cited in text.

4. The main purpose of a review article is to critically analyze the literature, identify the gap, and provide directions for future research. All of these are missing in this paper.

Reviewer 3 Report

This study aims to perform a bibliometric analysis of the top 50 most cited articles on the special Olympics. The study is quite interesting with the appropriate and thorough methods. It is rather well written; however, some comments and raised issues should be addressed.

Introduction:

·       The authors should provide additional rationale for this study. “There is an urgent need to analyse…” is not quite a scientific approach. Further in the manuscript, I’ve released the importance of this type of research, which should already be evident in the Introduction.

Materials and Methods:

·       Overall, the methods are clear and well written.

Results and Discussion:

·       In my humble opinion, the increase in publications in many sports science disciplines has been observed in the past 5-10 years. I don’t think this is only because of a growing interest in the Special Olympics movement. The authors should be cautious in concluding in such an overly confident manner. Please find some additional proof for this. Also, consider writing to leave the space for other options. For example: “These increases in publication numbers COULD BE A RESULT OF a consistency...”.

·       I understand this is bibliographic analysis. However, there are too many results and less discussion. Where possible, the authors should elaborate on some of the results. For example, why are the most productive authors from the USA, Canada, UK, and European countries? Or why do sixteen agencies fund only one publication each?

Conclusion:

·       The authors should provide a more meaningful conclusion. In particular, where are these gaps in the literature? By reading the discussion, I did not find any of these.

Overall, this manuscript requires more discussion, elaboration, and precise conclusions. Instead, the authors focused too much on sheer analysis (which was done excellent). However, casual readers and other researchers interested in this topic often seek more detailed information and explanations of some phenomena.